# Energetic robustness to large scale structural fluctuations in a photosynthetic supercomplex

Dvir Harris[1,5], Hila Toporik[2,3,4,5], Gabriela S. Schlau-Cohen [1] ✉ & Yuval Mazor [2,3] ✉

Photosynthetic organisms transport and convert solar energy with near-unity quantum efficiency using large protein supercomplexes held in flexible membranes. The individual proteins position chlorophylls to tight tolerances considered critical for fast and efficient energy transfer. The variability in protein organization within the supercomplexes, and how efficiency is maintained despite variability, had been unresolved. Here, we report on structural heterogeneity in the 2-MDa cyanobacterial PSI-IsiA photosynthetic supercomplex observed using Cryo-EM, revealing large-scale variances in the positions of IsiA relative to PSI. Single-molecule measurements found efficient IsiA-to-PSI energy transfer across all conformations, along with signatures of transiently decoupled IsiA. Structure based calculations showed that rapid IsiA-to-PSI energy transfer is always maintained, and even increases by three-fold in rare conformations via IsiA-specific chls. We postulate that antennae design mitigates structural fluctuations, providing a mechanism for robust energy transfer in the flexible membrane.

Oxygenic photosynthesis provides atmospheric oxygen while consuming carbon dioxide to enable the existence of complex life forms[1,2]. The light reactions of oxygenic photosynthesis include the photosystem I (PSI) and photosystem II (PSII) reaction centres (RCs) that perform light-induced charge separation. Most photosynthetic organisms utilize peripheral antennae associated with PSI and PSII to increase and control light absorption. In vivo, multiple antennae proteins are connected to each photosystem core, forming large photosynthetic units comprising of hundreds or thousands of light harvesting chlorophylls (chls) and typically organized in proteins within flexible cellular membranes. Within each antenna-photosystem supercomplex, light energy migrates between antenna chls in a process called excitation energy transfer (EET) before reaching the core photosystem[3,4]. EET is exquisitely sensitive to the distances, energies, and orientations of neighboring chls, which are tuned via the coordinating protein environment. In spite of their large compositional variability and the inherit sensitivity to fluctuations of tightly packed chls arrays, photosynthetic organisms can achieve a quantum efficiency close to unity in the dynamic and fluctuating cellular membrane[5–9].

Cyanobacteria use two types of antenna systems: (i) membrane bound iron-stressed induced protein A (IsiA) and prochlorophyte chlorophyll a/b-binding (Pcb) protein families and (ii) phycobilisomes, massive water-soluble pigment-protein complexes[10,11]. The IsiA/Pcb families contain six trans-membrane helices and shares its overall architecture with two core photosystem II subunits, CP47 and CP43. This protein fold is one of the main building blocks for oxygenic photosynthesis, as the same fold is found in both of the main PSI subunits PsaA and PsaB. The IsiA antenna is expressed under high-light or oxidative stress, but is most known for its expression under iron-

[1]Department of Chemistry, Massachusetts Institute of Technology, 77 Massachusetts Avenue, Cambridge, MA 02139, USA. [2]Biodesign Institute, School of Molecular Sciences, Arizona State University, Tempe, AZ 85801, USA. [3]School of Molecular Sciences, Arizona State University, Tempe, AZ, USA. [4]Faculty of Agriculture, Food and Environment, The Hebrew University of Jerusalem, Rehovot, Israel. [5]These authors contributed equally: Dvir Harris, Hila Toporik. ✉e-mail: gssc@mit.edu; ymazor@asu.edu

limited growth. Iron limitation is common in aqueous habitats, particularly in the Earth's oceans owing to its low solubility in water[12]. Lower levels of bio-available iron leads to lower amounts of iron-containing PSI complexes within the thylakoid membrane[10]. It has been proposed that IsiA serves as a light harvesting antenna, binding to PSI to increase its activity, compensating for the decrease in PSI levels in the membrane. In addition, it has been suggested that IsiA acts as an excess energy quencher and/or stores chls for later insertion into photosystems[10,13]. However, the precise function(s), and their underlying mechanisms, remain undetermined. Depending on the growth conditions, IsiA forms many assemblies with and without PSI, which have been hypothesized to correspond to the different functional roles[14,15]. Short-term iron starvation leads to the formation of PSI-IsiA complexes, while prolonged stress results in the formation of some IsiA-only assemblies[16,17]. The structure of supercomplexes with $PSI_3$-$IsiA_{18}$ stoichiometry have been determined[18–22]. However, the corresponding photophysical properties of these supercomplexes have been challenging to assign with open questions about the level of energetic connectivity between IsiA and PSI as well as the roles of specific pigments in the pathways of energy transfer[14,15,23,24]. One major challenge in interpreting the dynamics arises from measuring an ensemble of supercomplexes, as the nature of the heterogeneity within and between the complexes as well as the impact of the heterogeneous assemblies on function had not yet been determined. Previous ultrafast spectroscopic investigations of PSI-IsiA complexes suggest that in this organization, IsiA serves as an efficient PSI antenna, funneling excitation energy in a single-digit ps timescale[25,26], leading to very efficient trapping at PSI[27–30]. Investigations of IsiA-only assemblies showed a wide range of fluorescence decay timescales ranging from pico- to nanoseconds, where the faster components became dominant in large assemblies[14,31–33]. Consistent with this picture, a similar trend has been long established for another chlorophyll-binding antenna protein, Light harvesting complex II (LHCII) from higher plants. LHCII functions as both an antenna and a quencher[34,35], and reorganizes into arrays under quenching conditions[36–38]. A similar correlation between the nature of the assembly and its function may be present in IsiA. We previously observed that in the cyanobacterial PSI trimer each PSI monomer can adopt a range of positions relative to the other monomers[39]. In the case of the PSI-IsiA complex, it was not clear if these positional fluctuations persist and how they propagate into the IsiA antennae.

In this work, we examine these questions by collecting a high-resolution cryogenic electron microscopy (Cryo-EM) dataset of PSI-IsiA from *Synechocystis* sp. PCC 6803. Accounting for the heterogeneity in the data set reveals a surprisingly large conformational space consisting of changes in IsiA-PSI orientations and connectivity. Probing individual PSI-IsiA complexes using single-molecule spectroscopy reveals that efficient IsiA to PSI energy transfer dominates along with reversible changes in the PSI-IsiA connectivity. These changes likely reflect transient separation of IsiA from PSI, which may serve as the first step in formation of the other assemblies when needed but have easily reversibility to otherwise maintain intact supercomplexes. Calculations suggest robust efficiency is achieved through the presence of multiple pathways for IsiA to PSI energy transfer strategically positioned at some terminal emitters in IsiA. We suggest that this robustness to large scale structural heterogeneity is a significant driver for the evolution and shape of antenna–photosystem interaction surfaces. These findings have broad implications for other membrane-bound antenna systems, and provide a design principle for robust, bio-inspired solar energy devices.

## Results

### PSI-IsiA structural flexibility

The structure of $PSI_3$-$IsiA_{18}$ from Synechocystis sp. PCC 6803 was determined using Cryo-EM to an overall resolution below 2.8 Å

(Fig. 1a, b, Supplementary Fig. 1). The structure is consistent with previous reports with the canonical organization of the $PSI_3$-$IsiA_{18}$ supercomplex, a trimeric PSI surrounded by a ring-forming 18 IsiA protomers[18–20]. Each IsiA protomer coordinates 17 chls and 4 carotenoid molecules. Because of the three-fold pseudo-symmetry of $PSI_3$-$IsiA_{18}$, a comparable and significant improvement in resolution was achieved by using C3 symmetry or C3 symmetry expansion (Supplementary Fig. 1). Each of the IsiA-IsiA interaction interfaces occurs over a large solvent excluded surface area of ~1600 Å². In contrast, the IsiA-PSI interactions are much weaker, occurring in <1% of their total solvent accessible surfaces. These interactions take place mostly over two patches: (1) the PsaF/J subunits with an average area of around 1100 Å²; and (2) the PsaK subunit with an area of around 300 Å² (Fig. 1d). These same regions are also where the closest chls connecting PSI to IsiA are found (Fig. 1f, g). Overall, a picture of stronger IsiA-IsiA interactions and labile PSI-IsiA interactions emerges from both the physical and energy transfer perspectives.

We carried out multibody refinement in relion[40] to explore the heterogeneity in the PSI-IsiA data set suggested by the differences in resolution of PSI and the IsiA ring. Multibody refinement incorporates heterogeneity in a dataset as a collection of independent rigid bodies defined by user provided masks. Mask selection defines how the heterogeneity within the structural model is visualized. To identify the masks that best capture the heterogeneity of the supercomplex, we compared the resolution in one PSI monomer and one IsiA hexamer across different masking approaches. Supplementary Fig. 3A displays the different masks used, ranging from splitting the PSI-IsiA complex as thirds $PSI_1$-$IsiA_6$ (i.e. a PSI monomer rigidly coupled to an IsiA hexamer) through treating IsiA as a separate ring of 18 subunits. It can clearly be seen that masking PSI as a monomer and IsiA as a hexamer resulted in the best reconstruction with the highest resolution of 3.26 Å and 2.58 Å (at FSC = 0.143) for IsiA and PSI, respectively, compared to other approaches (Supplementary Fig. 3C–D). Especially noticeable is the fact that treating PSI-IsiA as thirds is inferior to all other treatments, with a 2.8 Å and 3.7 Å resolution (at FSC = 0.143) for the PSI and IsiA domains, respectively. This strongly support the notion that substantial levels of heterogeneity exist between PSI and the IsiA hexamer. Smaller masks (IsiA trimers and dimers) did not result in improved resolution compared to the IsiA hexamer with the current dataset, and so the structural heterogeneity associated with the organization of smaller IsiA assemblies was not resolved using Cryo-EM.

The $IsiA_{18}$ ring can be split into hexamers at six different positions relative to the interface with the PSI monomer, we compared the resolution of the reconstructed maps from the six masks (Supplementary Fig. 4). The masks that maximizes the interface with a single PSI monomer (abcdef and fabcde in Supplementary Fig. 4) resulted in the highest $IsiA_6$ map resolution (3.25 Å at FSC = 0.143), while the mask in which the $IsiA_6$ interacted with two PSI monomers to similar extent resulted in the lowest resolution (3.4 Å at FSC = 0.143, Supplementary Fig. 4). This suggests that positional heterogeneity in the ring is maximized at the hexamer junctions, around the IsiA 'f' position. In agreement with this interpretation, the interaction surfaces of IsiA 'f' with its neighboring IsiA monomers, either 'e' within the hexamer or with 'a⁺¹' between the hexamers, are the smallest. The solvent excluded surface area between IsiA 'e' and IsiA 'f' is profoundly smaller compared to all other monomer-monomer interactions (986 ± 34 Å², Fig. 1e), suggesting that this association is particularly labile. Using the 'abcdef' mask which maximizes the PSI interface and allowed us to consider energy transfer to a single PSI monomer, we characterized the heterogeneity in the particle dataset using principal component analysis as implemented in relion[40,41]. We carried out refinement runs on each third of the complex using separate masks for the PSI monomer and the IsiA hexamer. Since the dataset was expanded using C3 symmetry, highly similar results are expected across each third and we used the different thirds to estimate the variability of the refinement itself.

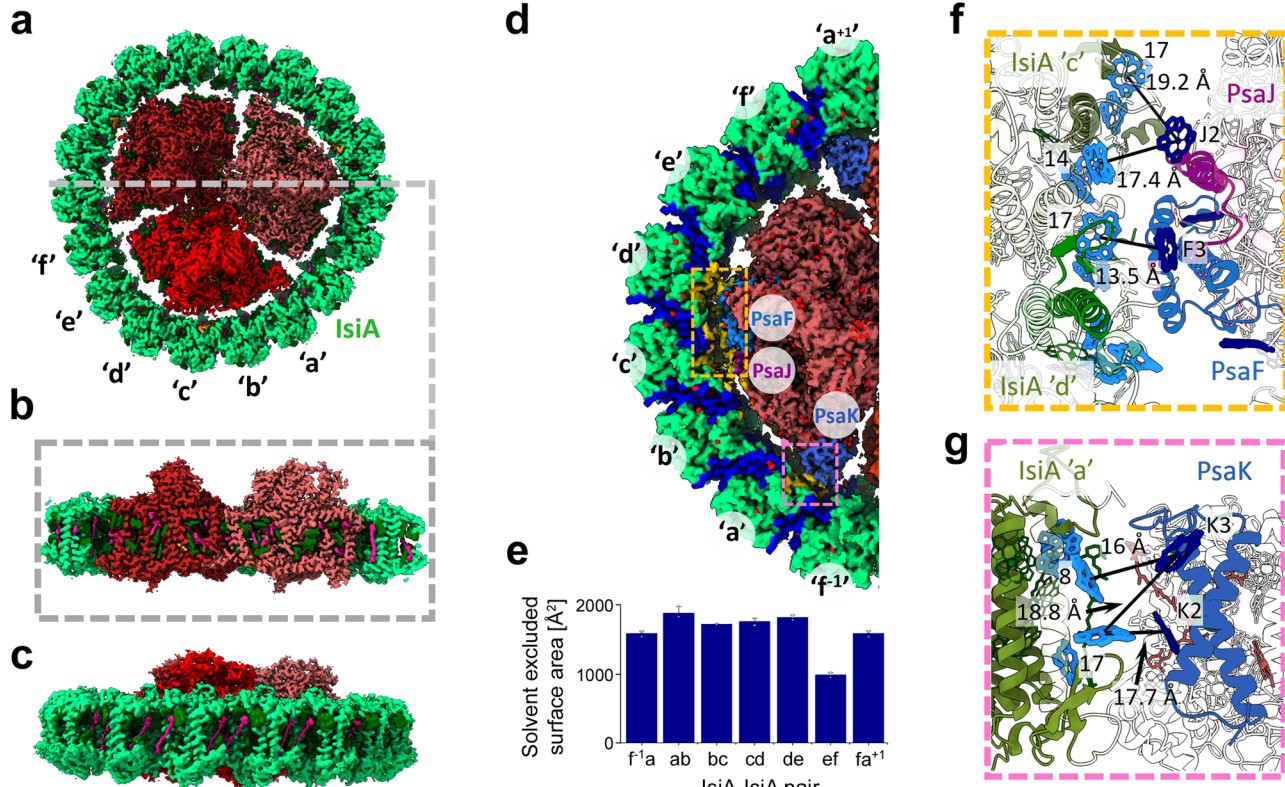

**Fig. 1 | Cryo-EM structure of PSI-IsiA. a** $PSI_3$-$IsiA_{18}$ viewed from the stromal face of the membrane with the three PSI monomers in shades of red and the dodecameric ring of IsiA in green. IsiA hexameric positions are denoted in black as 'a' to 'f'. **b** Cross section through PSI-IsiA at the gray dashed line in (**a**). Chls are shown in green, carotenes in pink and lipids in orange. **c** $PSI_3$-$IsiA_{18}$ viewed from the membrane side. **d** Region of $PSI_3$-$IsiA_{18}$ viewed from the stromal face of the membrane. The solvent excluded surfaces between individual IsiA subunits are in blue. Surfaces excluded due to IsiA-PSI interactions are in orange and are only detected between PsaF/J, PsaK on PSI and IsiA. **e** Values of each IsiA-IsiA solvent excluded surface area within the $IsiA_6$ and between $IsiA_6$ and its neighboring $IsiA_6$'s. Standard deviations are shown as error bars and are calculated from each third of the complex ($n = 3$). Source data are provided as a Source Data file. **f** The PsaJ/F−IsiA interaction surface viewed from the lumenal side of the membrane. IsiA chls 14 and 17 are closest to PSI in all IsiA subunits, which is the closest distances in the ring found in this interaction surface. **g** PsaK−IsiA interaction area viewed from the membrane plane. The shortest distances between PSI chls and IsiA chls are indicated. The location of IsiA chl 17 at the middle of the membrane is also shown.

Across all three thirds, the first six principal components (PC) explained more than 80% of the variance in the dataset (Fig. 2a and Supplementary Fig. 5). The particle distributions along each of the six PC axis do not show resolved peaks, suggesting no distinct PSI-IsiA conformations are present (Fig. 2b and Supplementary Fig. 5B, E and H). To visualize the shape of these PC's in real space we used the structures of the 9th and 91st percentile (in pink and blue respectively) in the particle population along each PC axis and superimposed the maps of the PSI monomer to highlight the positional range of $IsiA_6$ relative to PSI. The results for the first four PC's are shown in Fig. 2 and reveal high levels of heterogeneity in the $IsiA_6$ position. PC1 consists of an $IsiA_6$ tilting motion in and out of the membrane plane with IsiA'd' serving as a hinge. At the hexamer edges, IsiA monomers are translated by 14 and 24 Å. PC2 displays a lateral rotation of $IsiA_6$ relative to PSI with all IsiA monomers in the $IsiA_6$ experiencing a similar pattern of motion. As the diameter of an individual IsiA monomer is ~30 Å, this rotation means that, to a good approximation, monomers shift into the position of their neighbor along the PSI interface. All possible positions around PSI are sampled as part of this PC (albeit at different population frequencies). PC3 shows rotational translation of the IsiA antennae that causes a small diagonal displacement with respect to PSI. Lastly, PC4 demonstrates a profound vertical rotation, resulting in a very wide range of IsiA antennae motion in and out of membrane plane. In the case of PC1, the magnitude of the translations can be affected by the rigid mask (meaning that at the mask extremities translations are

larger). However, this effect is not observed for PC's 2-4. To examine the ability of $PSI_3$-$IsiA_{18}$ to maintain structural integrity in the face of these translations, we reconstructed plausible configurations at representative positions along the PC's. $PSI_3$-$IsiA_{18}$ were reconstructed from their thirds using different PC's by minimizing clashes and maximizing the overlap between adjacent $PSI_1$-$IsiA_6$ (These overlaps are included as part of the soft edges of each mask, Supplementary Fig. 6). This led to the identification of three PC combinations which could be matched along their lengths to form reasonable supercomplexes along all population percentiles (Supplementary Fig. 6 and Supplementary Movies 1–3). These combinations visualize some of the structural heterogeneity within $PSI_3$-$IsiA_{18}$.

## PSI-IsiA photophysical variability

Cryo-EM analysis probes frozen samples, making it challenging to distinguish between static and dynamic structural variation. We turned to fluorescence spectroscopy to distinguish between those possibilities and characterize the impact of the variation on the photophysics of $PSI_3$-$IsiA_{18}$. The ensemble absorption and emission spectra are shown in Fig. 3a, and are in agreement with previously reported values for the PSI-IsiA supercomplex[23,25]. The fluorescence lifetime data exhibited multi-exponential kinetics (Supplementary Fig. 7B), which most commonly arise from multiple types of emitters within the measured sample, consistent with the heterogeneous structures observed in Cryo-EM. The decay curves were best fitted with a tri-exponential function dominated by a component faster than the

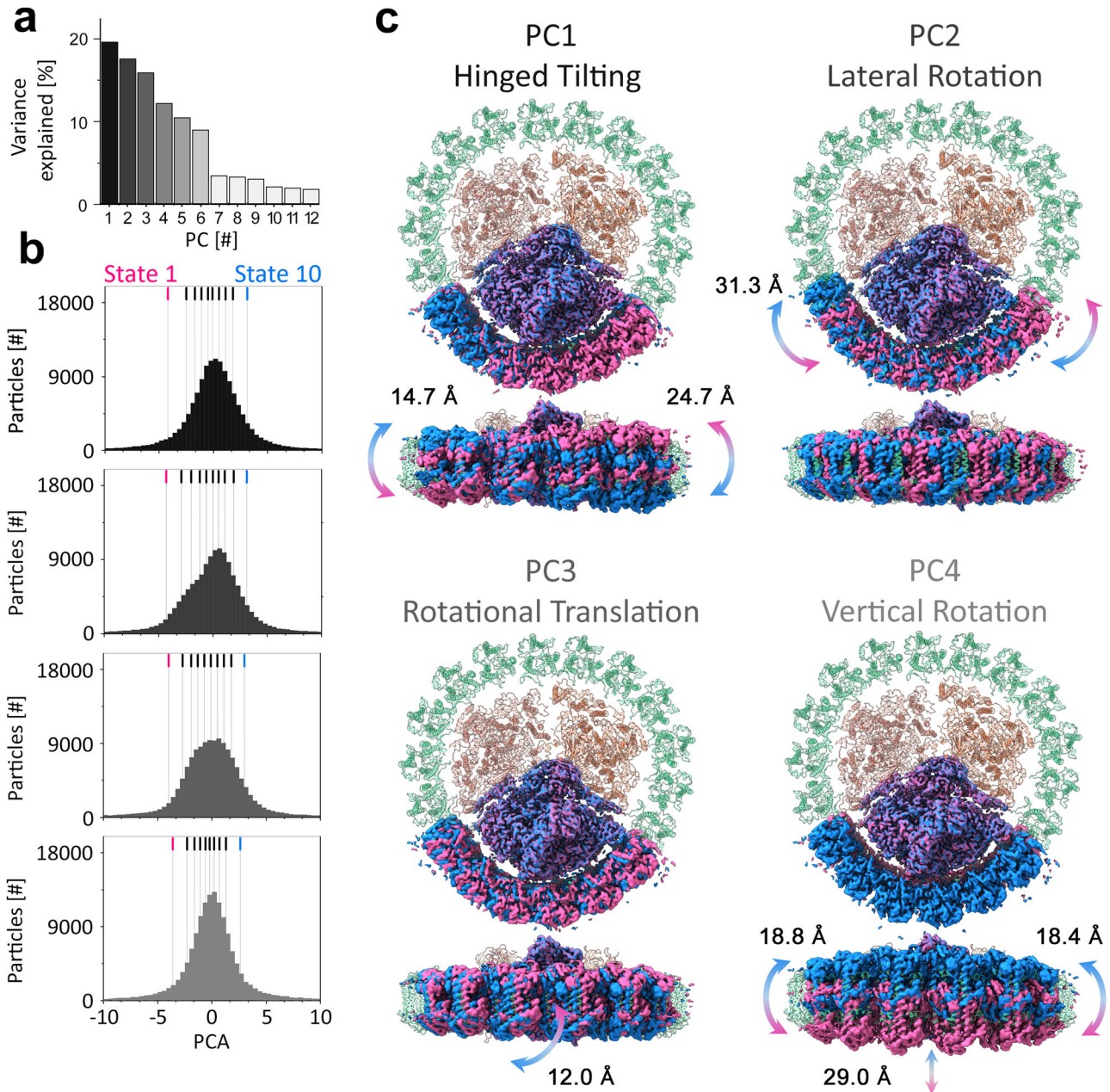

**Fig. 2 | Large scale structural heterogeneity in PSI-IsiA. a** The variance explained by each of the 12 principal components in the data. More then 80% of the variance is explained by the first six PCs. **b** Representative particle distributions along the four most dominant PC's show continuous heterogeneity. Vertical lines mark 9% population intervals and the 9 (state 1) and 91st (state 10) percentiles are marked in pink and blue respectively. **c** PCs visualized in real space as maps of PSI and IsiA$_6$ shifted to the state 1 or state 10 positions along PCs and superposed on PSI. The extent of translation on specific locations is indicated. Type of motion for each PC is defined in the title above the maps. Title of each motion is color-coded to match the PCs variance contributions shown in panel (**a**) and their respective histograms shown in panel (**b**). Source data for panels (**a**) and (**b**) are provided as a Source Data file.

temporal resolution of the measurement (<45 ps, 96%) along with two additional components (88 ± 3 ps, 3%, 2.4 ± 0.04 ns, 1%). The fast component likely arises from a combination of the ~1–10 ps timescale of IsiA to PSI energy transfer, followed by trapping by PSI on a ~30 ps timescale.[23,26,42] The presence of slow components suggests decreased energetic connectivity between IsiA and PSI in ~4% of the sample. This can result from a minor subpopulation of stable, less connected supercomplexes or from dynamic behavior of the majority of supercomplexes. The slowest component is two orders of magnitude slower than the trapping time[43]. This nanosecond decay is similar to other chlorophyll-containing antenna proteins[44–46]. Thus, the slow component likely arises from emission from isolated IsiA that is partially or fully energetically disconnected from the rest of the PSI$_3$-IsiA$_{18}$ supercomplex.

We investigated the heterogeneity within the PSI$_3$-IsiA$_{18}$ population by using single-molecule fluorescence spectroscopy to determine the distribution of photophysical behaviors. Representative fluorescence intensity-lifetime traces from individual PSI$_3$-IsiA$_{18}$ supercomplexes are shown in Fig. 3b and in Supplementary Fig. 8, where the fluorescence exhibits rapid transitions between intensity levels often accompanied by changes in the lifetimes. The lifetimes of individual supercomplexes revealed a mixture of mono-exponential decays faster than the measurement temporal resolution (<400 ps; Fig. 3b, c— light red) and bi-exponential decays with a similar fast component and

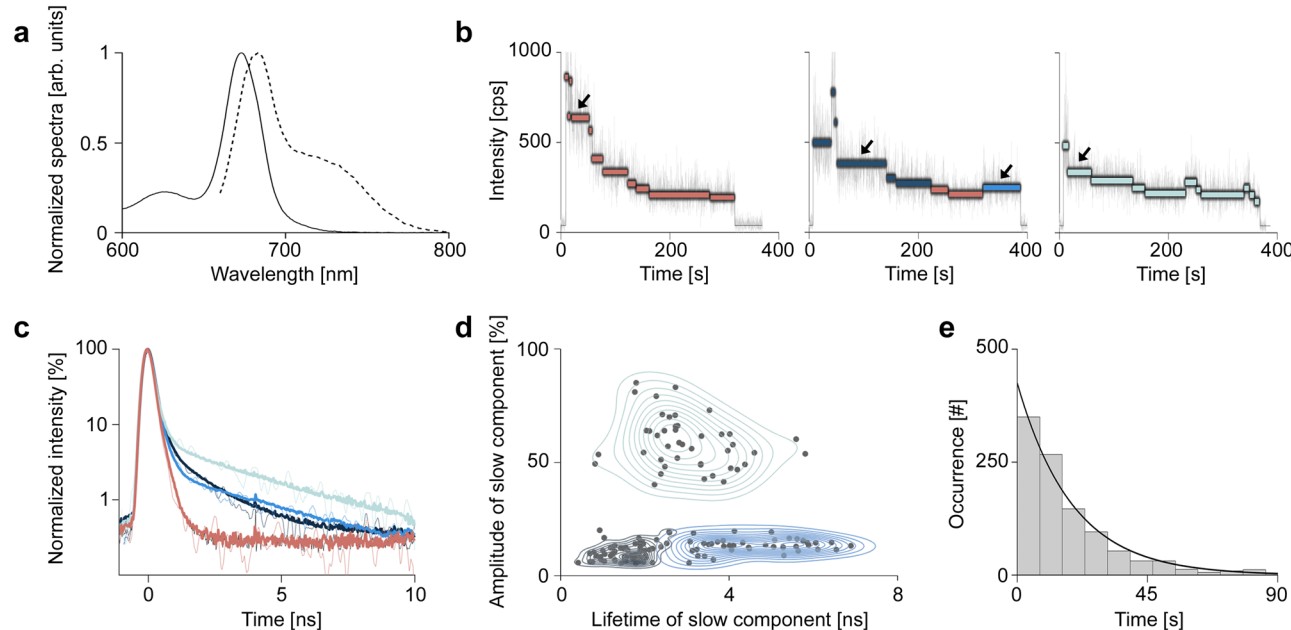

**Fig. 3 | Single-molecule spectroscopy reveals heterogeneity in PSI-IsiA energetic connectivity. a** Steady state, room-temperature absorption (solid line) and emission (dashed line) spectra of bulk PSI-IsiA. **b** Representative single-molecule fluorescence intensity-lifetime time traces for `continuously coupled' (left), `transiently coupled' (middle) and `partially coupled' (right) PSI-IsiA complexes. Mono-exponential states are shown in light red while bi-exponential states are shown in either medium and dark blue (center) or cyan (right). Fluorescence lifetime decays of states indicated by arrows are shown in (**c**). **c** Fluorescence decay curves (thin lines) with corresponding fits (thick lines) for representative intensity levels indicated with arrows in (**b**) and shown in the corresponding colors. IRF is also shown (black, ~400 ps) and overlaps with the light red curve. **d** Plot of amplitude and lifetime of the slow component for intensity levels that exhibit bi-exponential fluorescence decay curves. The two clusters with an amplitude of ~10%, indicated as (I), correspond to the transiently coupled population and are shown with dark and medium blue contour lines. The cluster with an amplitude of ~50% indicated as (II), corresponds to the partially coupled population is shown with cyan contour lines. **e** Histogram of dwell times from the single-molecule intensity traces for PSI-IsiA supercomplexes with fit (purple). Source data for all panels are provided as a Source Data file.

a slower component (>1 ns; Fig. 3b, c—medium and dark blue, cyan). These photophysical states were observed in three types of particles: (i) continuously coupled, having only mono-exponential states (Fig. 3b, left); (ii) transiently coupled, having interleaving mono- and bi-exponential states (Fig. 3b, center); and (iii) partially coupled, having only bi-exponential states (Fig. 3b, right). As in the ensemble data, the rapid decay component identified in all supercomplexes reflects trapping by PSI either directly or following rapid transfer from IsiA and the slow decay component reflects emission from disconnected IsiA. Free chls would not be visible in the single-molecule measurement, and so the observation of the slow component in the single-molecule data strongly supports its assignment to disconnected IsiA, resolving the discrepancy in the literature[14,15,23]. The bi-exponential decays indicate the presence of supercomplexes with lower and/or heterogeneous energetic connectivity. The protein composition of the subpopulations is the same, as the PSI₃-IsiA₁₈ complexes were isolated in high purity[18]. Therefore, the differences in energetic connectivity likely arise from structural heterogeneity in the organization of the proteins.

To characterize the subpopulations with heterogeneous energetic connectivity in more detail, the timescales and amplitudes of the slow component for all bi-exponential decays were used to construct a scatter plot (Fig. 3d). Three features are visible in the distribution. For two of the features (indicated as (I) in Fig. 3d), the amplitude of the slow component is low with a median of 11.2% (Supplementary Fig. 9C and Fig. 3d). One of these features has a median lifetime for the slow component of 1.77 ns (Supplementary Fig. 9A and Fig. 3d, dark blue), while the other has a median of 4.78 ns Supplementary Fig. 9A and Fig. 3d, medium blue). The third feature (indicated as (II) in Fig. 3d) has a median lifetime of 2.9 ns (Supplementary Fig. 9C and Fig. 3d, cyan) with a median amplitude of 57.6% (Supplementary Fig. 9D and Fig. 3d,

cyan). The amplitude of the slower component approximates the proportion of antennae subunits with low energetic connectivity. Thus, the higher amplitude of the slow component in population II suggests a higher content of weakly coupled antenna proteins. The dynamics of the bi-exponential subpopulations were examined using the transitions between types of emissive states. Remarkably, population I switched between bi-exponential and mono-exponential decays (Fig. 3b, center) whereas population II exhibited bi-exponential decays for the entire observation time (Fig. 3b, right). The lower amplitude of the slow component in population I is consistent with the lower content of weakly coupled antennae, suggesting that they are closer to the highly coupled state.

In summary, three distinct, single molecule decay patterns were observed: 60% of the complexes showed fast mono-exponential decay at all intensity levels (Fig. 3b, left). 36% of the complexes showed fast mono-exponential decay intensity levels interleaved with bi-exponential decay levels (Fig. 3b, middle; population I in Fig. 3d, dark and medium blue). Lastly, 4% of the complexes showed bi-exponential decay at all intensity levels (Fig. 3b, right; population II in Fig. 3d, cyan). Thus, the 4% of energetic disconnectivity observed in the ensemble decay curves (Supplementary Fig. 7B), which is consistent with previous work showing highly efficient trapping in PSI-IsiA supercomplexes[27–30], actually arises from 40% of the complexes, which show permanent or transient energetic disconnectivity. To characterize the overall dynamics of the supercomplexes, the switching between different emissive states was also quantified. The dwell times, which are the duration of constant fluorescence intensity and lifetime values, were calculated from all the single-molecule traces. The extracted dwell times were then used to construct the histogram shown in Fig. 3e. A single exponential fit of the dwell times yielded a timescale of $\tau = 19.1$ s, consistent with slow, large-scale structural

fluctuations. The time constant of the different levels of the PSI$_3$-IsiA$_{18}$ supercomplex is much longer than an individual antenna protein, which is usually a few seconds[44–47]. This may stem from the slower timescale of inter-protein reorganization as compared to intra-protein conformational changes. Collectively, these results indicate that the PSI$_3$-IsiA$_{18}$ complexes maintain high levels of energetic connectivity with a substantial subpopulation that can reversibly disconnect from PSI.

### PSI-IsiA energy transfer pathways

Next, we turned to theoretical calculations based on the experimental structures to better understand the effect of the positional flexibility on energy transfer. We modeled IsiA-to-PSI energy transfer for each structures at 9% population percentiles along all the PC's (Fig. 2a). The rate of energy transfer was calculated using Förster resonance energy transfer (FRET) theory. The FRET rate was calculated for each IsiA$_{chl}$-PSI$_{chl}$ pair separated by <25 Å for each structure (Supplementary Table 2)[48,49]. The total rates, which are the sums of all the pairwise rates, are plotted in Fig. 4a and Supplementary Fig. 11A, bottom. The consensus structure gave rise to a rate of ~2 $ps^{-1}$, in agreement with the literature[26,42]. Surprisingly, the IsiA to PSI transfer rate increased up to three-fold for conformations at the tails of the PC distributions (Fig. 4a, c and d and Supplementary Table 2). This effect was observed in four of the six analyzed PCs (Fig. 4a, Supplementary Fig. 11, bottom). The two PCs that remained relatively flat throughout the population (3 and 5, Supplementary Fig. 11) show relatively small translations in real space (Supplementary Fig. 5). These results suggest that rare conformations of PSI-IsiA have the fastest energetic connectivity, and correspondingly higher energy transfer efficiency as compared to the dominant one.

IsiA contains four unique chls absent from CP43, the homologous PSII core antenna. We previously suggested that two of these chls (numbered 8 and 17) play the role of terminal emitters of IsiA, mainly based on their proximity to PSI[18]. Bar plots in Fig. 4a and Supplementary Fig. 11 (top), show the relative contributions of these chls to the overall IsiA to PSI energy transfer rates. These two chls, which account for only ~10% of total IsiA chl content, contribute over 50% of the total energy transfer rate to PSI for all structures along all PCs (Supplementary Fig. 11A, top). Moreover, these chls are especially critical at the tails of conformations, where their relative and absolute contributions increase in the four PC's (PC1, PC2, PC4, PC6) that show the largest real space translation (Supplementary Figs. 11, 12). This suggests that these chls not only play a general role in mediating the energetic connectivity of IsiA to PSI, but also provide energetic robustness at the most extreme architectures. In contrast, the contributions from another suggested terminal emitter of IsiA, chl 14, to the overall transfer rate was small across all PC's (Supplementary Fig. 12, center). This suggests that this chl does not play a critical role in maintaining strong and robust connectivity between IsiA and PSI and is more important for IsiA to IsiA transfer.

Two of the main transfer sites between IsiA and PSI are located near the PSI subunits PsaK and PsaJ/F (Fig. 1f, g). Both sites contain PSI chls, namely K3 and J3 respectively, proximal to the Chl 8–Chl 17 pair of different IsiA monomers. Figure 4c, d shows how the transfer rates change across the first PC at these two sites, where the sites display maximum efficiency at opposite extremities of PC1. Supplementary Fig. 12 and Supplementary Table 2 indicate that chl 17 is the major donor for energy transfer from IsiA to PSI with chl 8 also contributing to the process, especially upon vertical rotations (PC4 and PC6), as expected based on its vertical gap from chl 17. This pathway multiplicity provided by the two sites serves a mechanism to cope with the structural fluctuations.

### Discussion

PSI possesses the fastest trapping rates in all of photosynthesis. Even in very large complexes or ones with loosely bound antennae systems,

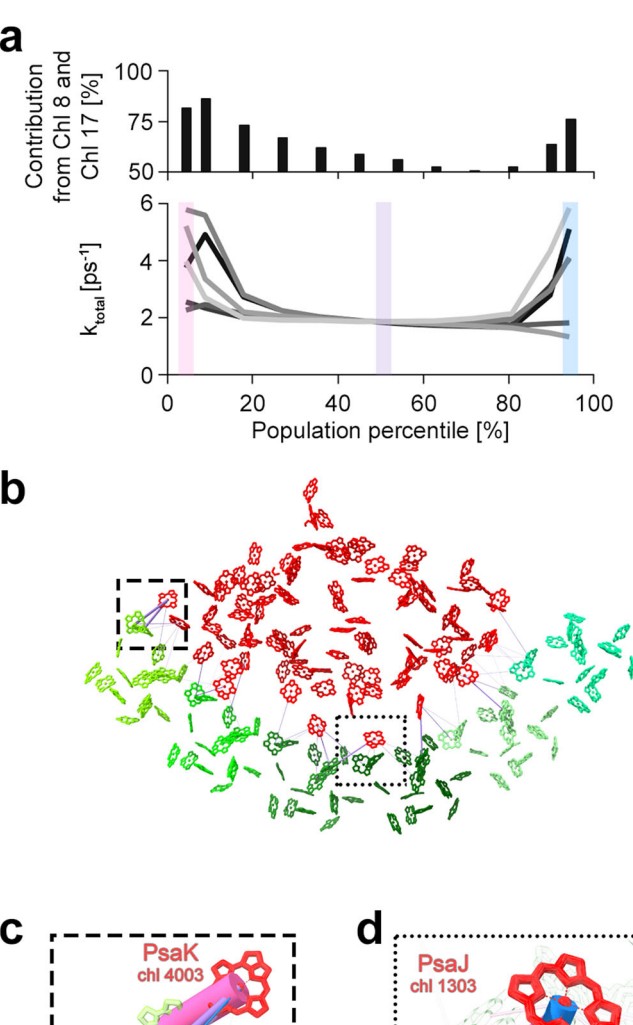

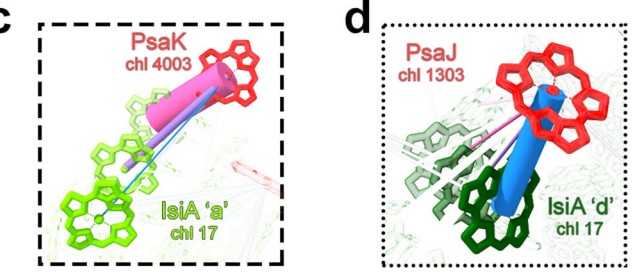

**Fig. 4 | Excitation energy transfer pathway is conformation-dependent. a** Bottom - For each principal component (PC1-6, from dark to light gray), the sum of Förster rates from every IsiA and PSI chls pair closer than 25 Angström is plotted in PSI-IsiA models representing different population percentiles. Source data are provided as a Source Data file. Top−Contribution from Chls 8 and 17 to the overall rate (for PC1). In all states, more than 50% of total rate was attributed to contributions from these two chls alone. **b** The consensus conformation (indicated by the purple shade in panel (**a**)) was depicted with its respective IsiA-PSI Förster resonance energy transfer (FRET) pairs. This model is virtually identical to the structure of PSI-IsiA. Panels (**c**) and (**d**) show the change in key chl-chl configurations (for PC1) at two main PSI-IsiA energy transfer sites, close to PsaK and PsaJ, respectively. Progressing from one end of the PC range (pink rate line) through the consensus conformation (purple rate line) to the other end of the PC (blue rate line), these segments were highlighted in panel (**a**), bottom, by shaded rectangles in the same colors. For clarity, other objects are mostly transparent. The thickness of each line is proportional to the FRET rate between both chls. All chls at the interface were accounted for when summing the rate (as indicated by the lines connecting the chls pairs in (**b**)).

the quantum efficiency of the entire system remains high with correspondingly fast trapping rates[27–30]. Given its typical fast trapping rate, the extent of movement that was previously observed in low resolution[50] and now determined in high-resolution in PSI$_3$-IsiA$_{18}$ between the antennae and core is extraordinary, with 20−30 Å

translations for individual IsiA subunits at specific locations. It has been recently shown that vitrification of CryoEM grids may introduce small narrowing of ensemble heterogeneity, due to various thermodynamic effects[51]. These effects can (and probably do) decrease molecular heterogeneity at cryogenic temperatures compared to room temperature. However, the magnitude of the structural changes discuss here is much larger and according to our dwell time analysis equilibrates over much longer time scales than those involved in the cooling process (seconds vs. nanoseconds, Fig. 3e). Therefore it is likely that additional structural changes associated with the observed heterogeneity (involving protein and ligand movements) will be affected by the vitrification process, but the slow, large scale, heterogeneity we describe is affected to a smaller degree by vitrification.

One possible explanation for the observed heterogeneity in $PSI_3$-$IsiA_{18}$ is the flexible nature of the thylakoid membrane. The overall structure of the thylakoid membranes in cyanobacteria is rather flat, but displays local flexibility in the form of different degrees of curvature[52]. The total membrane surface occupied by $PSI_3$-$IsiA_{18}$ is ~85 $nm^2$. Under low iron conditions, $PSI_3$-$IsiA_{18}$ complex and additional IsiA assemblies are expected to occupy the majority of the thylakoid membrane surface[53]. Therefore, the properties of the $PSI_3$-$IsiA_{18}$ particles should reflect the nature of the entire membrane system. We suggest that the structural elements that maintain fast transfer rates in the face of structural heterogeneity in $PSI_3$-$IsiA_{18}$ reflect adaptation to sustain highly efficient EET in the face of membrane flexibility. The nature of the structural heterogeneity suggest that changes in transfer rates are minimized over all observed PC's. For example, we did not observe side-to-side "movements" of the IsiA ring in relation to PSI, which would impact the chls distances maximally. Instead, most of the variability is nearly orthogonal to the shortest PSI-IsiA chls connections, which is the simplest way to minimize changes in transfer rates in supercomplexes.

Interestingly, our analysis showed that the most efficiently connected configurations were not the most common in the population, but rather at the distribution tails. This property is observed in most PC's and is mostly attributed to IsiA specific chls (numbered 8 and 17) located close to the membrane center and proximal to PSI chls (Supplementary Fig. 10). This is in contrast to Chl 14 which is also positioned at the PSI-IsiA interface and is proposed to be a terminal emitter in both CP43 and IsiA[54]. Overall, these outcomes suggest that multiplicity of energy transfer pathways, as well as donors located close to the center of membrane, evolved to counter structural fluctuations within a noisy membrane environment and minimize quantum yield loss in light of the unexpected large motions. Similarly, our spectroscopic results show that fast energy transfer is maintained in all particles, along with spectroscopic signatures of lower connectivity. The approach we used to model structural heterogeneity does not treat individual IsiA monomers or assemblies smaller than $PSI_1$-$IsiA_6$ complexes due to low signal intensity. While it was possible to reconstruct some of the structural heterogeneity at the $PSI_3$-$IsiA_{18}$ supercomplexes level (Supplementary Fig. 6), it is likely that individual IsiA subunits do become uncoupled in certain conformations and this can account for the appearance of a transient bi-exponential state (Fig. 3b, center; and population I in Fig. 3d). The reversible nature of the transition into a conformation with a bi-exponential lifetime decay suggests that $PSI_3$-$IsiA_{18}$ does undergo slow structural transitions at room temperature with functional consequences. In the particles with bi-exponential decays, the slow component likely arises from disconnected IsiA subunit(s), assigned through their characteristic nanosecond lifetime decay[14,47,55]. Our single-molecule results elucidate that previously observed low-amplitude, long-lifetime fluorescence signal coming from PSI-IsiA samples does not arise from proportionally low amounts of residual IsiA contaminates, but rather from PSI-IsiA complexes that exhibit transient or permanent components from uncoupled antenna. The surprising ubiquity of this behavior may reflect the dynamic

reorganization of IsiA into different assemblies in response to changes in growth or nutrient conditions. As our CryoEM data show fully assembled $PSI_3$-$IsiA_{18}$ complexes, these would most likely correspond to the dominant subpopulation with full energetic connectivity. Given that the three subpopulations have the same protein composition, they are inherently interchangeable, albeit not necessarily within the timescale of the measurement. The transitions in and out of different subpopulations may reflect a dynamic nature for the structural organization that facilitates PSI-IsiA assembly and disassembly. The IsiA monomer at the 'f' position may be involved in these emissions because of its relatively small interaction surfaces with its neighboring proteins. In the transiently decoupled states IsiA 'f' may remain proximal to the supercomplex, enabling re-binding and restoration of the strongly coupled $PSI_3$-$IsiA_{18}$ feature. This would be in good agreement with the relatively slow mobility of IsiA in membranes[56]. It is unlikely that IsiA 'a' and/or 'd' monomers would detach from the complex due to their relatively large interaction surfaces with their neighboring partners and PSI. Consistently, strong energetic connectivity was observed in all the supercomplexes in our spectroscopic measurements and theoretical calculations indicated these monomers serve as the main conduits of energy transfer to PSI. It is possible that the process of PSI-IsiA complex destabilization at the 'f' position(s) is the first step towards either the formation of IsiA-only complexes upon prolonged iron stress, or the restoration of free PSI complexes upon reestablishing replete nutrient conditions. The presence of constant fluctuations of the IsiA subunits away from PSI under all conditions may contribute to rapid reorganization of the supercomplexes upon changes in growth conditions. A few seemingly contradictory roles have been proposed for IsiA, namely that it serves as both a PSI antenna and an energy quencher. Its antenna role is already suggested by the presence of an energy gap between IsiA and PSI, which allows energy to be funneled downhill to PSI. In this report, we focused on its role as a PSI antenna, but it is quite possible that the structural fluctuations we describe here and their modification of IsiA photophysical properties occur in IsiA-only assemblies. These fluctuations may facilitate the transition between organizations or even the formation of quenching centers.

In the broader context, our findings highlight that resistance to the effects of heterogeneity induced by flexible membrane environment is one of the driving forces in the evolution of IsiA as an antenna (directing the physical location of IsiA terminal emitters), and likely play a role in the shaping of other photosynthetic antennae. This approach may also be true for other membrane-bound protein complexes which, like PSI-IsiA, occupy large membrane surfaces and maintain their performance in constantly fluctuating surroundings. In conclusion, we found large and unexpected levels of structural and photophysical heterogeneity in $PSI_3$-$IsiA_{18}$, one of the largest and most prevalent light harvesting systems on earth. We suggest that energetic robustness is maintained thanks to specific elements in the chromophoric network evolved to counteract the heterogeneity imposed by biological membranes. Our findings, therefore, inform on new designing principles for bio-inspired solar energy converters operating under noisy, highly fluctuating environments.

## Methods

### Culture conditions

Cyanobacteria were cultured in glass bottles in 10 liters batches using BG11 medium supplemented with 12*ng*$ml^{-1}$ ferric ammonium citrate and 5 mM glucose at 30 °C and bubbled with air. Light was supplied from a light-emitting diode array (Fluence RAY) at very low intensity (~15 μE).

### PSI-IsiA purification

Between 20 and 40 liters of culture were harvested by centrifugation and washed once with STN1 buffer (30 mM Tricine-NaOH pH 8, 15 mM

NaCl, 0.4M sucrose). Cells were resuspended in STN1 and broken with two cycles at 30,000 psi in a cell disruptor (Constant Systems Ltd). The lysate was cleared by centrifugation in a F20-12 × 50 LEX rotor (Thermo Scientific) for 10 min at 18,500 × $g$. Membranes in the supernatant were pelleted using ultracentrifugation (Ti70 rotor, 149,000 × $g$, 2 h) and resuspended in STN2 buffer (30 mM Tricine-NaOH pH 8, 100 mM NaCl, 0.4 M sucrose). After resuspension in STN2, the membranes were incubated on ice for 30 min then collected again (Ti70 rotor, 149,000 × $g$, 2 h) and resuspended in STN1. n-Dodecyl $\beta$-D-maltoside (DDM, Glycon) was added to the membranes at a 10:1 DDM-chlorophyll ratio. The suspension was gently mixed manually a few times then incubated on ice for 30 min. After solubilization, the insoluble material was discarded using ultracentrifugation (Ti70 rotor, 149,000 × $g$, 30 min). The solubilized membranes were loaded onto a diethylamino ethanol column (Toyopearl DEAE-650C). The complexes were eluted using a linear NaCl gradient (15-500mM NaCl) in 30 mM Tricine-NaOH pH 8, 0.2% DDM. Dark green fractions were collected and precipitated using 6% PEG3350 (Hampton Research). After centrifugation in a F20-12 × 50 LEX rotor for 5 min at 3200 × $g$, the green precipitate was resuspended in 30 mM Tricine-NaOH pH 8, 75mM NaCl with 0.05% DDM and loaded onto a 12–60% sucrose density gradient prepared with the same buffer. Following centrifugation (Beckman SW40 rotor, 243,500 × $g$, 16 h), the appropriate green band was precipitated using 10% PEG3350 (Hampton Research). After centrifugation in Eppendorf tabletop for 5 min at 11,000 × $g$, the green precipitate was resuspended in 30 mM Tricine-NaOH pH 8, 75 mM NaCl with 0.05% DDM and loaded onto a 12–60% sucrose density gradient prepared with the same buffer. Following centrifugation (Beckman SW60 rotor, 422,500 × $g$, 4 h), the appropriate green band was collected and used for subsequent experiments.

### Sample preparation for single-particle cryo-EM analysis
The PSI-IsiA band from the sucrose gradient was collected and the buffer was exchanged to 30 mM Tricine-NaOH pH 8, 75 mM NaCl, and 0.02% DDM using a Sephadex G-50 gel filtration column. The PSI-IsiA complex was concentrated using a spin column (Spin-X UF 100k, Corning) to 1.5 mg chlorophyll ml$^{-1}$. A 3-μl drop of the PSI-IsiA complex was applied on to a holey carbon grid (C-flat 1.2/1.3 Cu 400-mesh grids, Protochips) after soaking the grid in buffer. The sample was vitrified by flash-plunging the grid into liquid ethane using an automated plunge freezer, a Vitrobot Mark IV (ThermoFisher/FEI) with a blotting time of 6 s. The grids were stored in liquid nitrogen before data acquisition.

### Data acquisition
The cryo-EM specimens were imaged on a Titan Krios transmission electron microscope (ThermoFisher/FEI). Electron images were recorded using serialEM[57] on a K2 Summit direct electron detect camera (Gatan) at super-resolution counting mode. The defocus was set to vary from −1 to −3 μm and the nominal magnification was ×47,600, corresponding to a super-resolution pixel size of 0.525 Å at the specimen level. The counting rate was adjusted to 7.37 e-/Å$^2$s. Total exposure time was 8s, accumulating to a dose of 59 e-/Å$^2$.

### Data processing
A flowchart describing data handling is shown in Supplementary Fig. 1. MotionCor2[58] was used to register the translation of each sub-frame, and the generated averages were Fourier-cropped to ×2 and dose-weighted. Contrast transfer function (CTF) parameters for each movie were determined using CTFFIND4[59]. Particles were picked using TOPAZ[60]. Relion3.1 was then utilized for subsequent data processing[61]. Unsupervised 2D classification was used to pick a set of 129,517 particles which were extracted from the original micrographs as boxes of 400 pixels (1.05 Å/px). This particle set yielded a 4.1 Å resolution 3D map of PSI-IsiA using the known structure as an initial model (filtered to a resolution of 60 Å). Multiple rounds of 2D classification without

alignment resulted in 88,759 particles that were subjected to 3D classification and yielded one major class containing 75,688 particles. Repeated rounds of particle filtering using 2D classification with tilt and per-particle CTF refinement[62] were carried out, resulting in 47,913 particles which yielded a 3.16 Å resolution map in C1. C3 expansion of this set resulted in a 2.84 Å map which was used as the starting point for multibody refinement[41]. Maps of individual PSI monomers and IsiA hexameres were sharpened using a value of −51 or −69 respectively using the postprocessing step in relion. The final PSI-IsiA map used for model building was assembled in Phenix[63] using the Combined-focused-map procedure. Local resolution was estimated using ResMap[64].

### Model building and refinement
The initial PSI model was taken from the previous *Synechocystis* PSI-IsiA (PDB: 6NWA)[18]. The model was docked into the map using UCSF Chimera[65]. Monomeric IsiA was rebuilt from subunit X in 6NWA using Coot[66]. The final model was refined against the Cryo-EM density map using phenix.real_space_refine[67]. Final model statistics are shown in Supplementary Table 1. PyMOL[68] and UCSF ChimeraX were used to generate all images[69].

### Modeling PSI-IsiA across PCs
For modeling conformations of PSI-IsiA, a PSI monomer and the corresponding IsiA hexamer were docked into the PC map independently as rigid bodies using ChimeraX.

### Ensemble spectroscopic measurements
Absorption and emission spectra of the PSI-IsiA sample were collected using an Epoch Microplate Spectrophotometer (BioTek) and a Cary Eclipse Fluorescence Spectrophotometer, respectively (Fig. 2a). Fluorescence lifetime measurements were performed using a super-continuum generated in a nonlinear photonic crystal fiber (Femto-White 800, NKT photonics) pumped by a Ti:sapphire oscillator (Mai Tai, Spectra Physics). The supercontinuum was then spectrally filtered to separate 630–655 nm with a band-pass filter (ET645/30x, Chroma). The emission was collected within 667–731 nm with a band-pass filter (ET700/75m, Chroma). The instrument response function (IRF) was measured to be ~90 ps (fwhm) using a scatter solution containing a 1:100 v:v mixture of HS-40 colloidal silica (Sigma-Aldrich) in tricine buffer. The fluorescence decay curves were fit by convolving a tri-exponential function with the measured IRF with a home-built MATLAB code.

### Single-molecule spectroscopic measurements
The PSI-IsiA sample was diluted to ~20 pM with 30 mM Tricine, 15 mM NaCl, 0.05% $\beta$-DDM, pH = 8 buffer. The sample was then immobilized on a glass coverslip by spincoating in 5% PVA. The coverslip was mounted on a piezoelectric stage (Mad City labs, Nano-LP100). The excitation source was generated by a tunable fiber laser (FemtoFiber pro, Toptica Photonics, 80 MHz repetition rate, 130 fs pulse duration, 610 nm, 4 nm full-width half maximum (fwhm)), passed through a pinhole, and directed into a home-built confocal microscope. The excitation was focused by an oil-immersion objective (UPLSA-PO100XO, Olympus, NA 1.4) onto the coverslip. The concentration of the sample resulted in ~4–5 complexes per 25 μm$^2$ scanned area. A piezoelectric stage was used to center the excitation laser on each complex and the emission was collected through the same objective and separated from the excitation using a dichroic (ZT647rdc, Chroma) and two band pass filters (ET690/120x and ET700/75m, Chroma; Supplementary Fig. 7A). The pulse energies were attenuated to 5100 nJ/cm$^2$ per pulse at the sample plane. Emission was detected by a silicon-based single photon counting avalanche photodiode (SPCM-AQRH, Excelitas Technologies). A time-correlated single-photon counting (TCSPC) module (Time Tagger 20, Swabian Instruments) was

used to record the arrival times for each detected photon. The instrument response function (IRF) was measured to be ~400 ps (fwhm).

## Single-molecule data analysis

The number of detected photons was binned at 100 ms resolution to generate fluorescence intensity traces (Fig. 2b). Periods of constant intensity (states) were identified by a change-point algorithm[70], and the arrival times for the photons during each period were histogrammed to generate fluorescence decay curves. The histograms were fit with either a single-exponential or a bi-exponential function convolved with the IRF with a separately measured background contribution using maximum likelihood estimation (MLE).

## Förster resonance energy transfer (FRET) rate calculation

We calculated the the Förster rate[48], similarly to ref. [49]. In short, the Förster energy transfer rate, $k_{DA}$, between a donor (D) and an acceptor molecule (A) is defined as $k_{DA} = C_{DA}\kappa^2/N^4 R_{DA}^6$. $C_{DA}$ is a factor calculated from the overlap integral between the donor emission and acceptor absorbance (taken from ref. [71]). $\kappa$ is the dipole orientation factor, defined as:

$$\kappa = (\hat{\mu}_D * \hat{\mu}_A - 3(\hat{\mu}_D * \hat{R}_{DA})(\hat{\mu}_A * \hat{R}_{DA})) \quad (1)$$

where $\hat{\mu}_A$ and $\hat{\mu}_D$ are the dipole unit vectors (taken as the vector between the $N_B$ and $N_D$ atoms of the specific chlorophyll molecule). N is the refractive index of the medium and $\hat{R}_{DA}$ is the distance between the two pigments (taken as the Mg-Mg distance between each pair of chlorophylls). For the sum of rate constants, only IsiA to PSI chl pairs separated by <25 Å were included. Calculations were performed using an R script (available from the author upon request).

## Reporting summary

Further information on research design is available in the Nature Portfolio Reporting Summary linked to this article.

# Data availability

The cryo-EM density maps generated in this study have been deposited in the Electron Microscopy Data Bank (EMDB) under accession code: EMD-26601. The atomic coordinates generated in this study have been deposited in the Protein Data Bank (PDB) under the accession code: 7UMH. The initial model used in this work was taken from PDB entry 6NWA. The R code used to calculate transfer rates is available from the authors upon request. Source data are provided with this paper.

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

## Acknowledgements

D.H. and G.S.S.-C. were primarily supported by the U.S. Department of Energy, Office of Science, Office of Basic Energy Sciences, Division of Chemical Sciences, Geosciences, and Biosciences under Award # DE-SC0018097 (to G.S.S.-C). Y.M. acknowledges the support by the National Science Foundation under Award No. 2034021 and U.S. Department of Energy, Office of Science, Office of Basic Energy Sciences, Division of Chemical Sciences, Geosciences, and Biosciences under Award # DE-SC0022956. D.H. would like to acknowledge the Yad Hanadiv (Rothschild) Foundation, the Zuckerman STEM Leadership Program, and the Israel Council for Higher Education (CHE) for their generous financial support.

## Author contributions

Y.M. and G.S.S.-C. designed research; D.H. and H.T. performed research; D.H., H.T., Y.M., and G.S.S.-C. analyzed data; and D.H., Y.M., and G.S.S.-C. wrote the manuscript.

## Competing interests

The authors declare no competing interests.
