## [Peer Review File · Nature Communications]

REVIEWER COMMENTS

Reviewer #1 (Remarks to the Author):

The manuscript by Harris et al. describes the structural heterogeneity and the robust energy transfer of PSI-IsiA supercomplex from *Synechocystis* sp. PCC 6803. The authors solved high-resolution structure of the PSI-IsiA through cryo-EM, performed multi-body refinement, and measured the fluorescence emission and lifetime of the complex. They showed that while PSI-IsiA undergoes large-scale structural fluctuations, the supercomplex efficiently transfers excitation energy from IsiA to the PSI core in all conformations. Two IsiA-specific chlorophylls, 8 and 17, are essential for the efficient EET.

PSI-IsiA structures from different cyanobacterial species were previously determined by several groups, whereas this work focuses on the heterogeneity of the supercomplex which is novel. However, during reading of the manuscript, some concerns about the technical issues of multi-body refinement arose.

1. The authors performed multi-body refinement using the data processed by C3 symmetry, is it better to use the data processed by C1 symmetry? Multi-body refinement is to describe continuous structural heterogeneity in the complex, however, slight conformational change may be averaged when applying C3 symmetry.

2. The strategy of masking seems to be important for the multi-body refinement. In this work, the authors tried different masking approaches, and chose the one with the best resolution of masking PSI as a monomer and IsiA as a hexamer. However, based on PSI-IsiA structures, the IsiA-IsiA interaction is strong, that it is difficult to imagine three IsiA hexamers moving independently in one complex. Would it be more reasonable to consider the 18 IsiAs as a whole and mask the IsiA ring separately from the three PSI monomers or one PSI trimer? Currently, the criterion for masking in this work is the reconstruction resolution, which may not be sufficient, as the resolution is not significantly different between reconstructions of different approaches, at least for some of them.

3. If the IsiA hexamer is masked, based on Figure 1E and Figure S3, the IsiA hexamer fabcde may be better than the hexamer abcdef.

Other comments

Line87, "The interfaces between these masks define the flexibility within the structural model". Flexibility is not defined by interfaces.

Line113, trimer -> hexamer

Line116, more then -> more than

Line120, "To visualize the shape of these PC's in real space we used the structures of the 9th and 91st percentile (in pink and blue respectively) in the particle population along each PC axis". Please explain why the authors chose the 9th and 91st percentile. Do they correspond to the first and tenth states in each component?

Line 145, "emission spectra are shown in Figures 3A. The emission spectrum contains the expected IsiA and PSI peaks, at 679 nm and 715 nm, respectively". I am not an expert in spectroscopy, but according to the cited references, room temperature fluorescence emission cannot distinguish between IsiA and PSI peak.

Fig. 1D, it's better to show PsaF, PsaJ and PsaK with different colors, right now, the positions of labels are not accurate.

Fig. 1G, this may be a luminal side view, as the PsaF and PsaJ are switched compared to those in Fig. 1D

Fig. 1G figure legend, IsiA chls 514 and 517 -> IsiA chls 14 and 17

Reviewer #2 (Remarks to the Author):

In this work, the authors report single particle cryo-EM and fluorescence spectroscopy measurements to make an important and physiologically relevant claim about the statistical heterogeneity of the IsiA-PSI complex of *Synechocystis* cyanobacteria. Their cryo-EM studies suggest that three different statistically significant deformations of the IsiA-PSI complex are seen. The observation of heterogeneity on the quaternary level of interprotein complexes is in itself physiologically relevant and should guide spectroscopic studies in the future and our understanding of energy flow dynamics in oxygenic photosynthesis. The authors further make the central claim by interpreting their single molecule fluorescence data that the energy transfer from IsiA to PSI is not affected by the overall complex adopting these different quaternary conformations. If correct, it is an extremely important and physiologically relevant claim that sheds light on the robustness of the fundamental process driving oxygenic photosynthesis: energy transfer between different protein complexes. Such a claim, if substantiated, is certainly worthy of publication in *Nature Communications*. I however find that their single molecule spectroscopy measurements and their analysis fail to substantiate this claim. Further relatively simple experiments are needed.

Detailed comments follow:

1. The authors should state the IRF of their instrument in the main text? They do mention that it is ~400ps in the SI, but a clear statement of this value is needed in the main text because the rates being compared are of this order of magnitude.

2. The authors do not cite two recent important works on the IsiA protein:

van der Weij-de Wit, C. D. et al. Fluorescence quenching of IsiA in early stage of iron deficiency and at cryogenic temperatures. *Biochim. Biophys. Acta* 1767, 1393–1400 (2007).

Chen, H.-Y. S., Niedzwiedzki, D. M., Bandyopadhyay, A., Biswas, S. & Pakrasi, H. B. A novel mode of photoprotection mediated by a cysteine residue in the chlorophyll protein IsiA. *MBio* 12, (2021).

From both works, the authors will see that the fluorescence of free IsiA is almost completely depleted by the first 400 ps (IRF). Therefore, the current single molecule spectroscopy measurements hardly confirm the high efficiency of transfer. It is totally possible that transfer is completely blocked in one of the conformations but the single molecule measurements are not picking it up because IsiA fluorescence is depleted.

On a similar note, the authors claim that the slow component corresponds to IsiA fluorescence and blocking of transfer to PSI because trapping in the PSI is fast. But this claim cannot be justified with the work's 400 ps IRF.

Similarly, in the discussion and the introduction, the following paper should also be cited for the fast PSI trapping rate in line 244:

Lee, Y., Gorka, M., Golbeck, J. H. & Anna, J. M. Ultrafast energy transfer involving the red chlorophylls of cyanobacterial photosystem I probed through two-dimensional electronic spectroscopy. *J. Am. Chem. Soc.* 140, 11631–11638 (2018).

Finally and most importantly, single molecule measurements are not required to make the central claim of this manuscript. The central claim can be easily verified by measuring the fluorescence quantum yield in an annihilation free excitation fluence of free IsiA, free PSI and the combined complex. These measurements may have already been reported in the literature, but if not, can be easily performed. If 95% quenching of fluorescence (after accounting for the absorption cross section of PSI on the membrane) is seen, then it is tantamount to saying that either closest connections between different proteins are not disturbed over conformational heterogeneity or that new connections that are as robust in the rarer conformers are formed thus resulting in highly efficient transport. (Note that while single molecule measurements may not be necessary, there is nothing wrong with them but the insights they provide should be corroborated by these simpler measurements especially given the uncertainty regarding IsiA fluorescence above.)

The central claim of this manuscript should therefore be substantiated with spectroscopic measurements of quantum yield (or reference to the QYs if they exist), or single molecule spectroscopy experiments should be performed with a much smaller IRF (smaller than 45 ps) to be able to actually observe transfer being blocked.

Of course, the other important observation from single molecule spectroscopy measurements is conformational switching between high-efficiency and not-so-high-efficiency energy transfer arrangements, and this observation can be a part of the manuscript I guess but is only expected to occur for the given the cryo-EM structures.

3. IsiA also suffers from an unclear understanding about its role in photosynthesis like the red Chl in PSI, Chl_z D2 Chls in PSII and overexpressed LH2-only membrane parts in low light purple bacteria. The authors should spend some time discussing what relevance their observations may have in ascribing a role to the IsiA18 cluster surrounding PSI. Is it photoprotective or is it just an antenna whose job is to efficiently funnel energy to PSI? Their observations seem to suggest it is the latter, but then it is surprising that IsiA and PSI are spectrally nearly degenerate, and no funnel-like structure is created to facilitate transfer. This discussion should further increase the relevance of the work.

Reviewer #3 (Remarks to the Author):

Harris et. al., used single particle analysis in combination with single molecule analysis to interrogate the structural heterogeneity and energetic coupling robustness between IsiA and the enclosed PSI trimer in IsiA-PSI super complex. Multibody refinement analysis of the complex indicated there are indeed several subpopulations of the complex, which was supported by single molecule decay analysis. The authors, however, didn't discuss how thermal factors could affect the overall results. Brock and Grubmüller have discussed the "Effects of cry-EM cooling on structural ensembles" Nature Communication, 2022. The authors need to include this reference and discuss how the thermal factors could affect the populations in cry-EM.

Another missing point in the manuscript is that how the heterogeneity observed in cryo-EM condition is related to the subpopulations observed in single molecule experiments. i.e., three decay patterns. Please discuss if the three subpopulations are interchangeable. In this case, a functionally inactive IsiA-PSI complex, as a control, could help clarify if the percentage pattern (60, 36, 4%) still stand. This also significantly helps define the robustness of the functional IsiA-PSI super complex presented in this manuscript.

Minor:

revise "ifs" in line 282 p14.

REVIEWER COMMENTS

Reviewer #1 (Remarks to the Author):

The manuscript by Harris et al. describes the structural heterogeneity and the robust energy transfer of PSI-IsiA supercomplex from *Synechocystis* sp. PCC 6803. The authors solved high-resolution structure of the PSI-IsiA through cryo-EM, performed multi-body refinement, and measured the fluorescence emission and lifetime of the complex. They showed that while PSI-IsiA undergoes large-scale structural fluctuations, the supercomplex efficiently transfers excitation energy from IsiA to the PSI core in all conformations. Two IsiA-specific chlorophylls, 8 and 17, are essential for the efficient EET.

PSI-IsiA structures from different cyanobacterial species were previously determined by several groups, whereas this work focuses on the heterogeneity of the supercomplex which is novel. However, during reading of the manuscript, some concerns about the technical issues of multi-body refinement arose.

We thank the reviewer for their effort and for acknowledging the novelty of our work. Our responses to all of the concerns raised are shown below in blue.

1. The authors performed multi-body refinement using the data processed by C3 symmetry, is it better to use the data processed by C1 symmetry? Multi-body refinement is to describe continuous structural heterogeneity in the complex, however, slight conformational change may be averaged when applying C3 symmetry.

On one hand, small (local) conformational changes, which are highly likely to accompany the structural changes we describe, may be masked when symmetry is applied. On the other hand, the better particle alignments obtained thanks to applying C3 symmetry may lead to better detection of such local variation. We attempted many classifications schemes using several masks in the complex and we detected no such changes (using both C1 or C3 symmetry).

To examine the effect of applying C3 symmetry on heterogeneity, we repeated the multibody analysis on our data without symmetry expansion. The results are shown in figure 1 of this response (next page). Although the results are not identical between the three thirds of the complex (this is ascribed to small variations in the refinement process itself, more on this subject below), they are highly similar and resemble the results obtained using C3 symmetry. In our opinion, the benefits of the better particle alignment obtained using symmetry expansion and the ability to look at the total variation in one third of PSI-IsiA, outweighs the theoretical benefits of preserving the C1 frame of reference. We agree that larger data sets may reveal additional changes in line with what the reviewer suggest.

Figure 1: Multibody analysis carried out on unexpanded particle stacks (using C1 symmetry). **A**, **C** and **E** show the variance explained by the different components and **B**, **D** and **F** show that extreme states in each PC, similarly to Supplementary figure 4.

2. The strategy of masking seems to be important for the multi-body refinement. In this work, the authors tried different masking approaches, and chose the one with the best resolution of masking PSI as a monomer and IsiA as a hexamer. However, based on PSI-IsiA structures, the IsiA-IsiA interaction is strong, that it is difficult to imagine three IsiA hexamers moving independently in one complex. Would it be more reasonable to consider the 18 IsiAs as a whole and mask the IsiA ring separately from the three PSI monomers or one PSI trimer? Currently, the criterion for masking in this work is the reconstruction resolution, which may not be sufficient, as the resolution is not significantly different between reconstructions of different approaches, at least for some of them.

We agree with the reviewer that the masking strategy is an important factor and to a certain extent determines the range of conformations we were able to describe. It is also a compromise between the signal to noise ratio of the data and our expectation for the optimal number of rigid bodies that are needed to account for the heterogeneity in the sample.

In figure 2 of this reply we include the multibody analysis carried out on using three PSI monomers and one complete IsiA ring.

Figure 2: PCA analysis of IsiA₁₈. **A.** The masks used in refinement (same as supplementary figure 2A, third option). **B.** The variance explained by each PC. **C.** The particles distribution and the shape of each PC (shown similarly to Supplementary figure 4).

As can be seen in figure 2 of this reply, the shape of some PC's (mainly the ones involving rotation, PC1 and PC3) can clearly be observed using an IsiA₁₈ mask. At the same time, using this mask means that the heterogeneity within the IsiA₁₈ ring itself is not considered, and based on the improved resolution using IsiA₆, this heterogeneity is significant. In addition to the improved resolution IsiA₆ based maps also resulted in better structural models, meaning that chemically correct features such as helix geometry and side chain conformations were improved. There is little doubt that additional levels of heterogeneity remain hidden, and using smaller masks can potentially expose this, however this remains outside our reach with the current data set.

3. If the IsiA hexamer is masked, based on Figure 1E and Figure S3, the IsiA hexamer fabcde may be better than the hexamer abcdef.

We attribute the FSC differences between the two hexamer masks (“abcdef” vs. “fabcde”; Black vs yellow traces in Supplementary figure 3B) to the refinement process itself. To verify this, we compared the average FSC from three runs across the entire PSI-IsiA (these runs were done on C3 expanded data). In figure 3 of this reply we show the average and SD of these runs for the two masking options.

Figure 3: The FSC differences between the ‘abcdef’ and the ‘fabcde’ hexamers masks falls within the variation of the refinement process itself. A. FSC curves averaged over the three possible positions of the two alternative hexamer masks (“abcdef” – yellow, “fabcde” – black) B. Zoom in on the FSC = 0.5 region. C. Zoom-in on the FSC = 0.143 region. Vertical bars are +/- 1 standard deviation in each resolution shell.

From resolution perspective the two masks are equivalent, we verified that the heterogeneity they uncover is also highly similar. Choosing the “abcdef” mask allows us to carry out the energy transfer analysis using a single PSI monomer and simplifies this analysis.

We are limited to considering a rigid IsiA hexamer in the current case, but given that we have shown that there is some level of heterogeneity in the conformations of the three PSI monomers, we think it also makes sense to choose the hexamer mask that maximizes that interface. We have indicated this in the revised version of the manuscript.

Other comments

Line 87, “The interfaces between these masks define the flexibility within the structural model”. Flexibility is not defined by interfaces.

This section of the manuscript now reads:

“Multibody refinement incorporates heterogeneity in a dataset as a collection of independent rigid bodies defined by user provided masks. Mask selection defines how the heterogeneity within the structural model is visualized. To identify the masks that best capture the heterogeneity of the supercomplex, we compared the resolution in one PSI monomer and one IsiA hexamer across different masking approaches.”

Line 113, trimer -> hexamer

Thanks, we have fixed the text.

Line 116, more then -> more than

Thanks, we have fixed the text.

Line 120, “To visualize the shape of these PC’s in real space we used the structures of the 9th and 91st percentile (in pink and blue respectively) in the particle population along each PC axis”. Please explain why the authors chose the 9th and 91st percentile. Do they correspond to the first and tenth states in each component?

These values are picked to obtain 10 internal positions along the PC axis. Yes, they do.

Line 145, “emission spectra are shown in Figures 3A. The emission spectrum contains the expected IsiA and PSI peaks, at 679 nm and 715 nm, respectively”. I am not an expert in spectroscopy, but according to the cited references, room temperature fluorescence emission cannot distinguish between IsiA and PSI peak.

Thanks, we have downgraded the assignments to merely say it is in good agreement with previous literature, now reading:

Results, line 154-156:

“The ensemble absorption and emission spectra are shown in Figure 3A, and are in agreement with previously reported values for the PSI-IsiA supercomplex [23, 25]”-

Fig. 1D, it’s better to show PsaF, PsaJ and PsaK with different colors, right now, the positions of labels are not accurate.

We did our best to improve this aspect of the figure. PsaF, J and K are in different colors.

Fig. 1G, this may be a luminal side view, as the PsaF and PsaJ are switched compared to those in Fig. 1D Corrected.

Fig. 1G figure legend, IsiA chls 514 and 517 -> IsiA chls 14 and 17 Corrected.

Reviewer #2 (Remarks to the Author):

In this work, the authors report single particle cryo-EM and fluorescence spectroscopy measurements to make an important and physiologically relevant claim about the statistical heterogeneity of the IsiA-PSI complex of *Synechocystis* cyanobacteria. Their cryo-EM studies suggest that three different statistically significant deformations of the IsiA-PSI complex are seen. The observation of heterogeneity on the quaternary level of interprotein complexes is in itself physiologically relevant and should guide spectroscopic studies in the future and our understanding of energy flow dynamics in oxygenic photosynthesis. The authors further make the central claim by interpreting their single molecule fluorescence data that the energy transfer from IsiA to PSI is not affected by the overall complex adopting these different quaternary conformations. If correct, it is an extremely important and physiologically relevant claim that sheds light on the robustness of the fundamental process driving oxygenic photosynthesis: energy transfer between different protein complexes. Such a claim, if substantiated, is certainly worthy of publication in *Nature Communications*. I however find that their single molecule spectroscopy measurements and their analysis fail to substantiate this claim. Further relatively simple experiments are needed.

We thank the reviewer for their time, effort and highlighting the importance and physiological relevance of our results.

Detailed comments follow:

1. The authors should state the IRF of their instrument in the main text? They do mention that it is ~400ps in the SI, but a clear statement of this value is needed in the main text because the rates being compared are of this order of magnitude.

We thank the reviewer for this suggestion and have added this information to the caption of Figure 3, which now reads:

“.....IRF is also shown (black; ~400 ps) and overlaps with the light red curve”.

2. The authors do not cite two recent important works on the IsiA protein:

van der Weij-de Wit, C. D. et al. Fluorescence quenching of IsiA in early stage of iron deficiency and at cryogenic temperatures. *Biochim. Biophys. Acta* 1767, 1393–1400 (2007).

Chen, H.-Y. S., Niedzwiedzki, D. M., Bandyopadhyay, A., Biswas, S. & Pakrasi, H. B. A novel mode of photoprotection mediated by a cysteine residue in the chlorophyll protein IsiA. *MBio* 12, (2021).

From both works, the authors will see that the fluorescence of free IsiA is almost completely depleted by the first 400 ps (IRF).

Therefore, the current single molecule spectroscopy measurements hardly confirm the high efficiency of transfer. It is totally possible that transfer is completely blocked in one of the conformations but the single molecule measurements are not picking it up because IsiA fluorescence is depleted.

On a similar note, the authors claim that the slow component corresponds to IsiA fluorescence and blocking of transfer to PSI because trapping in the PSI is fast. But this claim cannot be justified with the work's 400 ps IRF.

Similarly, in the discussion and the introduction, the following paper should also be cited for the fast PSI trapping rate in line 244:

Lee, Y., Gorka, M., Golbeck, J. H. & Anna, J. M. Ultrafast energy transfer involving the red chlorophylls of cyanobacterial photosystem I probed through two-dimensional electronic spectroscopy. *J. Am. Chem. Soc.* 140, 11631–11638 (2018).

Finally, and most importantly, single molecule measurements are not required to make the central claim of this manuscript. The central claim can be easily verified by measuring the fluorescence quantum yield in an annihilation free excitation fluence of free IsiA, free PSI and the combined complex. These measurements may have already been reported in the literature, but if not, can be easily performed. If 95% quenching of fluorescence (after accounting for the absorption cross section of PSI on the membrane) is seen, then it is tantamount to saying that either closest connections between different proteins are not disturbed over conformational heterogeneity or that new connections that are as robust in the rarer conformers are formed thus resulting in highly efficient transport. (Note that while single molecule measurements may not be necessary, there is nothing wrong with them but the insights they provide should be corroborated by these simpler measurements especially given the uncertainty regarding IsiA fluorescence above.)

The central claim of this manuscript should therefore be substantiated with spectroscopic measurements of quantum yield (or reference to the QYs if they exist), or single molecule spectroscopy experiments should be performed with a much smaller IRF (smaller than 45 ps) to be able to actually observe transfer being blocked.

Of course, the other important observation from single molecule spectroscopy measurements is conformational switching between high-efficiency and not-so-high-efficiency energy transfer arrangements, and this observation can be a part of the manuscript I guess but is only expected to occur for the given the cryo-EM structures.

First, we would like to thank the reviewer for bringing the recent papers on IsiA forward. We agree with the findings in these papers, which show a clear distinction between the fluorescence decay of high and low molecular weight forms of free IsiA. Fluorescence from high molecular weight free IsiA, *i.e.*, IsiA aggregates, is almost completely depleted in 400 ps, as highlighted by the Reviewer. In contrast, fluorescence from low molecular weight free IsiA, *i.e.*, unaggregated and/or monomeric IsiA, exhibits a much longer lifetime. Consistent with this picture, LHCII, one of the most well studied antenna proteins, exhibits much faster fluorescence decay in its aggregated form [Mullineaux et al., *BBA-Bioenergetics*, 1993; Tutkus et al., *JPPB Biology*, 2021]. LHCII and all other known antenna proteins exhibit nanosecond decay in their unaggregated forms [Palacios et al., *JPC B*, 2002; Son et al., *JACS*, 2021; Gruber et al., *Nanophotonics*, 2018].

Specifically, in Van der weij et al. (van der Weij-de Wit et al., *BBA-Bioenergetics*, 2007), IsiA aggregates were isolated similarly to an earlier paper from the same group (Ihalainen et al., *Biochemistry*, 2005). In their 2005 paper, the authors emphasize that their samples are aggregates: “Gel filtration experiments revealed that the isolated fraction consisted of very large complexes and was essentially free of monomeric IsiA” (Results section, paragraph 1, line 2). These aggregates are not present in our preparation of PSI-IsiA as established in our previous work (Toporik et al., *Nat Struc Mol Bio*, 2019).

In the case of Chen et al (Chen et al., *MBio*, 2021), free IsiA was isolated in a combination of monomeric and aggregated forms, and “slow” components of >2 ns with amplitudes of ~50 % existed in all samples (Figure 3 in (Chen et al., 2021)):

Similarly, slow components were also observed in free IsiA in an earlier report from the same group Figures 7 and 8 in (Chen et al., *BBA-Bioenergetics*, 2017).

Following these previous assignments, we assign the slow decay components to non-aggregated free IsiA. Consistent with this picture, in a separate manuscript currently under review, we isolated and characterized monomeric IsiA, which exhibited slow decay components:

Figure 4: Isolated IsiA monomers show fluorescence emission with long lifetime. (A) SDS-page shows purified IsiA monomers. (B) typical absorption (full line) and fluorescence emission (dotted line) spectra of IsiA monomers. (C) Time-correlated single photon counting (TCSPC) shows long lived fluorescence decay at the ensemble level, IRF (light gray, ~80 ps) is also shown. (D) typical single-molecule fluorescence emission trace showing distinct levels of intensity (left, black) and corresponding long lifetime for each defined state (right, red). (E) Exemplifying fluorescence decay of one state (medium gray) and its fit (red). IRF (light gray, ~400 ps) is also shown.

Thus, the presence of slow decay components serves as a spectroscopic signature of uncoupled and unaggregated IsiA. In contrast, the fast-trapping rate of PSI means that energetically coupled IsiA exhibit picosecond fluorescence decay. In our experiments, these two timescales are clearly resolvable and distinguishable.

We also agree with the Reviewer that other corroborating measurements are helpful to interpret and contextualize the results. Indeed, measurements that provide very similar information to that requested by the reviewer have already been carried out on PSI and PSI-IsiA from *Synechocystis* and from other cyanobacteria (Andrizhiyevskaya et al., *BBA-Bioenergetics*, 2004; Chauhan et al., *Biochemistry*, 2011; Gobets et al., *BBA-Bioenergetics*, 2001; Melkozernov et al., *Biochemistry*, 2003). They show rapid decay consistent with ~95% quenching in PSI and in PSI-IsiA, where additional slower decay components distinct

from that of free pigments with low amplitudes have also been observed for PSI-IsiA. In that sense, our conclusions are consistent with previous literature that PSI-IsiA operates with very high efficiency. Following the Reviewer's suggestion, we have referenced the previous measurements as well as clarified our assignments through the following additions:

Introduction, lines 52-61:

"Previous ultrafast spectroscopic investigations of PSI-IsiA complexes suggest that in this organization, IsiA serves as an efficient PSI antenna, funneling excitation energy in a single-digit ps timescale [25, 26], leading to very efficient trapping at PSI [27, 28, 29, 30]. Investigations of IsiA-only assemblies showed a wide range of fluorescence decay timescales ranging from pico- to nanoseconds, where the faster components became dominant in large assemblies [14, 31, 32, 33]. Consistent with this picture, a similar trend has been long established for another chlorophyll-binding antenna protein, Light harvesting complex II (LHCII) from higher plants. LHCII functions as both an antenna and a quencher [34,35], and reorganizes into arrays under quenching conditions [36, 37, 38]. A similar correlation between the nature of the assembly and its function may be present in IsiA."

Results, lines 166-168:

"The slowest component is two orders of magnitude slower than the trapping time [43]. This nanosecond decay is similar to other chlorophyll-containing antenna proteins [44, 45, 46]. Thus, the slow component likely arises from emission from isolated IsiA that is partially or fully energetically disconnected from the rest of the PSI₃-IsiA₁₈ supercomplex."

Results, lines 209-210:

"Which is consistent with previous work showing highly efficient trapping in PSI-IsiA supercomplexes [27, 28, 29, 30]."

Discussion, lines 304-305:

"In the particles with bi-exponential decays, the slow component likely arises from disconnected IsiA subunit(s), assigned through their characteristic nanosecond lifetime decay [14, 47, 55]."

In terms of the single-molecule measurements with improved temporal resolution, we agree it would be an exciting addition to explore the heterogeneity in the rapid decay component. However, single-molecule measurements with IRFs below 45 ps are currently impossible, as the high sensitivity required for single-molecule detection is inherently accompanied by a thicker photodiode active area, which increases the IRF.

Finally, we would like to highlight that, while previous work established an overall high efficiency of trapping, our results add that:

- (1) The high efficiency is maintained over the unexpectedly large range of structural heterogeneity observed in the cryoEM data.
- (2) The low efficiency component, although only 4%, remarkably arises from 40% of the complexes, which all exhibit static or transient channels of energetic disconnectivity. The ubiquity of these disconnected organizations may reflect the need for IsiA to dynamically reorganize into different assemblies depending on growth conditions.

These findings establish that single PSI-IsiA complexes experience large structural and energetic fluctuations, yet exhibit robust efficiency in the face of these fluctuations. We have clarified these key conclusions through the following additions:

Results, lines 208-211:

“Thus, the 4% of energetic disconnectivity observed in the ensemble decay curves (Supplementary figure 6B) actually arises from 40% of the complexes, which show permanent or transient energetic disconnectivity.”

Results, lines 291-294:

“Overall, these outcomes suggest that multiplicity of energy transfer pathways, as well as donors located close to the center of membrane, evolved to counter structural fluctuations within a noisy membrane environment and minimize quantum yield loss in light of the unexpected large motions.”

Discussion, lines 305-310:

“Our single-molecule results elucidate that previously observed low-amplitude, long-lifetime fluorescence signal coming from PSI-IsiA samples does not arise from proportionally low amounts of residual IsiA contaminates, but rather from PSI-IsiA complexes that exhibit transient or permanent components from uncoupled antenna. The surprising ubiquity of this behavior may reflect the dynamic reorganization of IsiA into different assemblies in response to changes in growth or nutrient conditions.”

3. IsiA also suffers from an unclear understanding about its role in photosynthesis like the red Chl in PSI, Chl_z D2 Chls in PSII and overexpressed LH2-only membrane parts in low light purple bacteria. The authors should spend some time discussing what relevance their observations may have in ascribing a role to the IsiA18 cluster surrounding PSI. Is it photoprotective or is it just an antenna whose job is to efficiently funnel energy to PSI? Their observations seem to suggest it is the latter, but then it is surprising that IsiA and PSI are spectrally nearly degenerate, and no funnel-like structure is created to facilitate transfer. This discussion should further increase the relevance of the work.

We agree with the reviewer that IsiA modalities are still unclear, and this indeed has been a main motivation for this study. We would like to note that in fact, there is an energetic funneling between IsiA and PSI, which is quite remarkable for a Chl a containing system. Specifically, it has been well-established that the Q_y absorption band of IsiA is blue-shifted (~10 nm) with respect to PSI Chls, thus facilitating the IsiA-to-PSI energy transfer directionality (Andrizhiyevskaya et al., *BBA-Bioenergetics*, 2004; Chen et al., *BBA-Bioenergetics*, 2017). Remarkably, this is in stark contrast to PSI antennae from higher plants, where the antennae are red-shifter compared to the PSI core, suggesting up-hill energy transfer (Croce et al., *BBA-Bioenergetics*, 2002; Wientjes et al., *Biophys J*, 2011). These spectral properties, along with our results as the Reviewer highlights, suggest that antenna is at least one of the functions of IsiA. We think it is likely that there is an assembly-dependent nature to its predominant function, in part owing to the aggregation-induced quenching discussed above.

Following the Reviewer's suggestion, we have added the following paragraph to our discussion addressing these points:

Discussion, lines 327-333:

“A few seemingly contradictory roles have been proposed for IsiA, namely that it serves as both a PSI antenna and an energy quencher. Its antenna role is already suggested by the presence of an energy gap between IsiA and PSI, which allows energy to be funneled downhill to PSI. In this report, we focused on its role as a PSI antenna, but it is quite possible that the structural fluctuations we describe here and their modification of IsiA photophysical properties occur in IsiA-only assemblies. These fluctuations may facilitate the transition between organizations or even the formation of quenching centers.”

Reviewer #3 (Remarks to the Author):

Harris et. al., used single particle analysis in combination with single molecule analysis to interrogate the structural heterogeneity and energetic coupling robustness between IsiA and the enclosed PSI trimer in IsiA-PSI super complex. Multibody refinement analysis of the complex indicated there are indeed several subpopulations of the complex, which was supported by single molecule decay analysis. The authors,

however, didn't discuss how thermal factors could affect the overall results. Brock and Grubmüller have discussed the "Effects of cryo-EM cooling on structural ensembles" Nature Communication, 2022. The authors need to include this reference and discuss how the thermal factors could affect the populations in cryo-EM.

We thank the reviewer for his time and for bringing this interesting paper forward. We think it is likely that the large displacements that IsiA assemblies experience with respect to PSI, include additional structural changes that are affected by narrowing occurring as part of the freezing process. It's difficult to directly compare the magnitude of the structural changes, but in general, *Bock et al* describe changes in the range of 1-6 Å (rmsf) across almost the entire population and all cooling rates, while the extent of structural changes across the particle population in our study can reach ~18 Å in some PC's (comparing IsiA from state 1 to 10 while superposing PSI).

In addition, the dwell time analysis from our single molecule studies describes a very slow process, on time scales of 10's of seconds which are not accessible to molecular dynamics and are unlikely to be affected by the rate of the cooling process which is completed by 128 ns in the longest runs considered in *Bock et al*. So, our conclusion is that an additional layer of heterogeneity may be associated with the continuous heterogeneity we are discussing in the current work. As stated above, these likely structural changes cannot be measured currently.

We have revised our discussion to include this text:

Discussion, lines 265-273:

"It has been recently shown that vitrification of CryoEM grids may introduce narrowing of ensemble heterogeneity, due to various thermodynamic effects [51]. These effects can (and probably do) increase molecular heterogeneity at room temperature compared cryogenic temperatures. However, the magnitude of the structural changes we discuss here is larger and according to our dwell time analysis equilibrates over much longer time scales than those involved in the cooling process (seconds Vs nanoseconds, Figure 3E). Therefore it is likely that additional structural changes associated with the observed heterogeneity (involving protein and ligands movements) will be affected by the vitrification process, but the slow, large scale, heterogeneity we describe is affected to a smaller degree by vitrification."

Another missing point in the manuscript is that how the heterogeneity observed in cryo-EM condition is related to the subpopulations observed in single molecule experiments. i.e., three decay patterns. Please discuss if the three subpopulations are interchangeable. In this case, a functionally inactive IsiA-PSI complex, as a control, could help clarify if the percentage pattern (60, 36, 4%) still stand. This also significantly helps define the robustness of the functional IsiA-PSI super complex presented in this manuscript.

The three single-molecule subpopulations have the same protein composition, as our measurements are performed on high purity PSI₃-IsiA₁₈ complexes. The dynamics of the single-molecule traces also show that the organizations are interchangeable, as we observe transitions between mono-exponential and bi-exponential behavior.

We cannot, however, assign a spectroscopic subpopulation to a specific PC, as all PC subpopulations show strong energetic connectivity.

As for perturbing the subpopulations percentages, it is challenging to imagine generating a functionally inactive PSI-IsiA complex as a control, given inactivity of this protein-pigment complex would mean fluorescent-dim assembly, making recording fluorescence signal impossible.

Following the Reviewer's suggestion, we have added the following text to our manuscript, discussing the interchangeable nature of the sample and the relation between the subpopulations and CryoEM data:

Results, lines 185-187:

“The protein composition of the subpopulations is the same, as the PSI_3 -LsiA₁₈ complexes were isolated in high purity [18]. Therefore, the differences in energetic connectivity likely arise from structural heterogeneity in the organization of the proteins.”

Discussion, lines 310-314:

“As our CryoEM data show fully-assembled PS_3 -LsiA₁₈ complexes, these would most likely correspond to the dominant subpopulation with full energetic connectivity.

Given that the three subpopulations have the same protein composition, they are inherently interchangeable, albeit not necessarily within the timescale of the measurement.

The transitions in and out of different subpopulations may reflect a dynamic nature for the structural organization that facilitates PSI -LsiA assembly and disassembly.”

Minor:

revise "ifs" in line 282 p14.

Thanks for the suggestion, we have changed this sentence which now reads:

“In the transiently decoupled states LsiA 'f' may remain proximal to the supercomplex, ~~enabling and retain ifs full pigment complement, to enable~~ re-binding and restoration of the strongly coupled PSI_3 -LsiA₁₈ feature”

REVIEWERS' COMMENTS

Reviewer #1 (Remarks to the Author):

The authors adequately responded to my questions. I have no additional comments.

Reviewer #2 (Remarks to the Author):

The authors have addressed my major concerns about aggregation (SI of prior paper) and fluorescence lifetime to abrogate the need for further experiments. I believe that their data supports the conclusion that the quaternary structure differs and that energy transfer appears robust to these differences.

The writing of the manuscript itself could be significantly improved to draw a more direct and impactful conclusion that might seed future work, but this is more of a stylistic suggestion.

As it stands, I would support publication.

Reviewer #3 (Remarks to the Author):

The authors have addressed my concerns/suggestions. Thanks.